# Relationships among social support, decision self-efficacy, and decision regret in colorectal chemotherapy cancer patients: A mediating model

Zhichao Duan[1☯‡], Ying Li[1☯‡], Gege Ren[1], Miao Tan[1], Fang Wu[2]*

**1** School of Nursing, Inner Mongolia Medical University, Hohhot, Inner Mongolia, China, **2** Department of Gastrointestinal Surgery, Affiliated Hospital of Inner Mongolia Medical University, Hohhot, Inner Mongolia, China

☯ These authors contributed equally to this work.
‡ Co-first authors.
* 15532354438@163.com

## Abstract

### Objective

This study aims to identify the factors associated with decision regret among colorectal cancer patients undergoing chemotherapy in China and to determine whether decision self-efficacy mediates the relationship between social support and decision regret.

### Methods

A cross-sectional study was conducted using convenience sampling with 243 colorectal cancer patients receiving chemotherapy. From 29 July 2025–2 October 2025., all participants were recruited from a tertiary hospital in Inner Mongolia, China. Data were collected through a sociodemographic and clinical questionnaire, the Social Support Scale, the Decision Self-Efficacy Scale, and the Three-Dimensional Decision Regret Scale. Non-parametric tests were employed to analyze associated factors, and mediation analysis was performed using SPSS and AMOS.

### Results

Educational attainment, occupational status, and monthly income were significantly associated with decision regret among patients. Further analysis revealed a negative association between social support and decision regret in individuals undergoing chemotherapy for colorectal cancer (β = −0.306, p < 0.001). Path estimates showed that social support was positively associated with decision self-efficacy (β = 0.471, p < 0.001), while decision self-efficacy was negatively associated with decision regret (β = −0.581, p < 0.001). Decision self-efficacy functioned as a mediator linking social

**Data availability statement:** All relevant data are within the manuscript and its Supporting Information files.

**Funding:** The author(s) received no specific funding for this work.

**Competing interests:** The authors have declared that no competing interests exist.

support to decision regret, producing an indirect effect of –0.273, which accounted for 47.24% of the total effect.

## Conclusion

Notable interactions were observed among social support, decision self-efficacy, and decision regret, with decision self-efficacy serving as the mediating mechanism. Clinicians are encouraged to prioritize strengthening both external social support and patients' internal decision self-efficacy, which were associated with lower decision regret and better quality of life. Additionally, particular attention should be directed toward individuals with lower income, limited employment, or lower educational attainment, as they demonstrate a heightened vulnerability to experiencing decision regret.

## 1 Introduction

Neoplasms of malignant nature that develop within the epithelial lining of the rectum or colon (termed colorectal cancer (CRC)) stand out as prevalent forms of cancer on a global scale [1]. Figures presented in the Global Cancer Statistics from the International Agency for Research on Cancer (IARC) [2], reveal about 1.926 million instances of initial CRC detection in 2022, which form 9.6% of overall cancer detections while fatalities from this disease make up 9.3% of cancer- induced mortalities. These figures underscore CRC as a major global public health burden. Currently, chemotherapy remains a key component of CRC treatment, aiming to eliminate residual disease after surgery and improve overall survival [3]. However, the complexity of chemotherapy regimens, potential toxicities, and the varied impacts of treatment options on patients' quality of life pose substantial challenges during decision-making [4]. CRC patients must decide not only whether to undergo chemotherapy, but also evaluate specific regimens, routes of administration, treatment schedules, and strategies for managing side effects [5]. When patients feel uncertain about the best option or perceive that they may have a suboptimal choice, decision regret can arise, often accompanied by dissatisfaction and self-blame [6].

Decision regret is a negative emotional state that typically arises when individuals reflect on past events or choices [7]. In healthcare, it involves not only regret regarding treatment outcomes but also evaluations of the decision-making process and option selection, and is regarded as a multidimensional experience [8,9]. In CRC care, decision regret is not only an important indicator for assessing decision quality, health status, and their effects on subsequent medical choices, and is also a key determinant of perceived outcomes and satisfaction with care of patients [10]. In a prospective cohort study conducted by Temitope G, approximately 31% of rectal cancer patients were classified as having high decision regret at 1-year of follow-up after surgery [11]. Mitigating decision regret is essential for individuals with CRC patients, as it improves their treatment experience enhances adherence to chemotherapy, and contributes to better overall quality of care.Previous studies have identified social support and decision self-efficacy as two major predictors of decision regret among cancer patients [12,13].

Decision self-efficacy (DSE) refers to the confidence and ability of patients with CRC to obtain and understand relevant information and to translate this information into concrete actions during chemotherapy decision-making. The focus is not on a single decision but on the overall sense of self-efficacy throughout the decision-making process, including the level of trust and perceived control of patients over the choices made [14]. Existing studies have shown that higher self-efficacy can reduce uncertainty and helplessness during decision-making and thereby lower the occurrence of decision regret [13]. However, Lee and Bryant-Lukosius reported that approximately 64% of patients with CRC have marked deficiencies in decision self-efficacy [15], which require urgent attention. social support is a key factor in enhancing decision self-efficacy. Studies have indicated that social support can strengthen cognitive appraisal and perceived control and thereby improve self-efficacy [16,17]. By improving social support, patients can receive greater informational, emotional, and psychological support during the decision-making process, which in turn increases confidence and autonomy [18].

According to the Stress-Buffering Model proposed by Cohen and Wills, Social support functions as a key buffering resource when individuals encounter stressful or uncertain situations and plays an important role in shaping psychological adaptation and behavioral outcomes [19,20]. This model conceptualizes social support as a multidimensional construct, including emotional, informational, and instrumental support including emotional, informational, and instrumental support. Previous studies have demonstrated the substantial impact of social support on medical decision-making processes and outcomes in patients. Insufficient social support may impair psychological buffering capacity in patients, increasing vulnerability to regret regarding prior treatment decisions [12,21]. However, the ways in which specific dimensions of social support differentially influence decision regret, and whether these effects are mediated by underlying psychological mechanisms, such as decision self-efficacy, remain unclear. Moreover, the Social Cognitive Theory (SCT) finds broad utilization in accounting for the onset of behavioral shifts via the interplay involving cognitive appraisal, behavioral capacity, and environmental support [22]. This theoretical model positions self-efficacy as a key element in the process of making decisions, embodying a person's belief regarding competence to execute fitting selections [23,24]. Data reveal that those diagnosed with CRC exhibiting greater demonstrate heightened propensity for proactive involvement in deliberations over chemotherapy [14]. In turn, higher decision self-efficacy reduces perceived barriers during the decision process, facilitates more active participation, and ultimately decreases the likelihood of decision regret [25]. Social support as an important external resource, can further enhance decision self-efficacy by providing emotional, informational, and instrumental assistance, thereby enabling patients to navigate uncertainty more confidently during treatment decision-making [13,21].

Although previous studies have examined the independent effects of social support and decision self-efficacy on decision regret, the existing evidence remains fragmented. Most research has focused on breast or prostate cancer, with limited attention to colorectal cancer patients undergoing chemotherapy— a population facing particularly complex and high-stakes treatment decisions [6]. Limited investigations have concurrently employed the Social Cognitive Theory and Stress-Buffering Model to elucidate mental processes via which social support shapes decision regret. Empirical substantiation for the intermediary function of decision self-efficacy remains lacking amid chemotherapeutic contexts for CRC. Such voids receive rectification through this effort's fusion of dual synergistic conceptual paradigms alongside structural equation modeling, which gauges immediate and mediated conduits tying social support to decision regret, thereby propelling the repository of prior scholarship. Fig 1 depicts the conceptual model used in this study.

## 2 Aims and hypotheses

This study aimed to identify factors associated with decision regret in colorectal cancer (CRC) patients and to examine the proposed mediating role of decision self-efficacy in the relationship between social support and regret. The study tested three hypotheses: (H1) There is a negative association between social support and decision regret; (H2) There is a negative association between decision self-efficacy and decision regret; and (H3) Decision self-efficacy mediates the association between social support and decision regret.

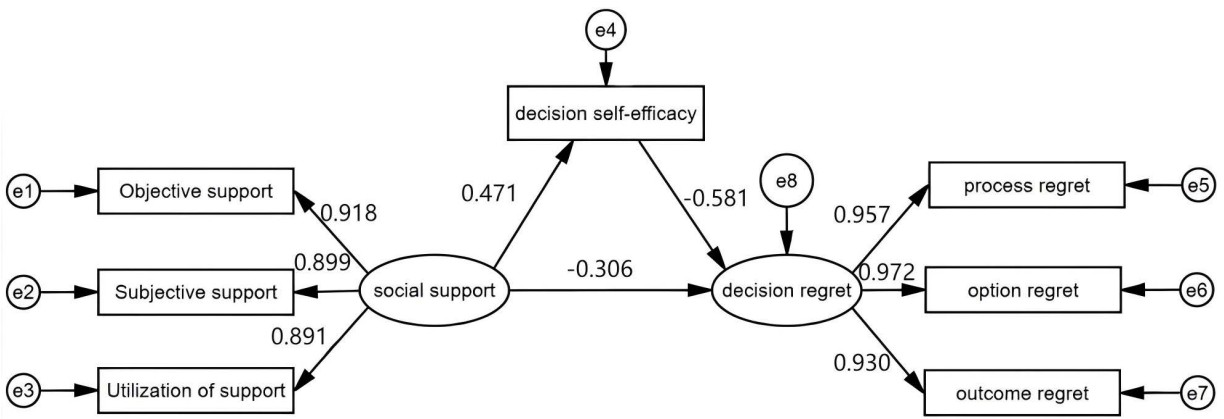

**Fig 1. Mediating model of social support, decision self-efficacy, and decision regret in colorectal cancer patients undergoing chemotherapy.**
This figure illustrates the structural equation model examining the mediating effect of decision self-efficacy on the association between social support and decision regret. Social support is represented by three observed indicators: objective support, subjective support, and utilization of support. Decision regret is represented by process regret, option regret, and outcome regret. Standardized path coefficients are shown along each pathway, with solid arrows indicating significant direct effects.

## 3 Methods

### 3.1 Study design

Data gathering in the current cross-sectional inquiry spanned the interval between 29 July 2025 and 2 October 2025. Observational findings underwent documentation aligned with directives from the Strengthening the Reporting of Observational Studies in Epidemiology (STROBE) framework.

### 3.2 Setting, participants, and procedure

Individuals receiving for CRC underwent enrollment through convenience sampling at the Oncology Department alongside the Chemotherapy Day within a tertiary care institution located in Inner Mongolia, China. Criteria permitting participation included: (1) CRC established via pathological examination; (2) having undergone a minimum of one chemotherapy regimen; (3) being at least 18 years old; (4) exhibiting lucid awareness; (5) possessing standard communicative skills; and (6) providing full dataset from evaluations. Criteria barring involvement consisted of the following:1) had a coexistence of other malignancies; (2) had severe hepatic or renal dysfunction; (3) were deemed by their attending physician to have poor compliance or refused to cooperate with the study.

 With the assistance of the head nurse in the oncology department,research team members informed participants of the purpose and significance of the study. Informed consent was secured from each participant prior to their willing enrollment in the investigation. Skilled personnel delivered and retrieved surveys through individualized interactions amid the inpatient units. Privacy safeguards dictated that every instrument underwent anonymous fulfillment, supplemented by on-the-spot guidance to rectify incomplete entries at retrieval. Investigators further conveyed to chosen individuals that remorse over prior therapeutic selections qualifies as a standard emotional response, urging candid disclosure of personal insights and sentiments linked to such selections within the instruments. Respondents encountering literacy barriers benefited from verbal recitation of the surveys by the investigative group, followed by transcription of their replies.

 All study variables were measured at a single time point during the patient's current chemotherapy visit. Participants completed the questionnaires before receiving their scheduled chemotherapy cycle on that day. Although patients were at different stages of chemotherapy, all measures were collected cross-sectionally and were not repeated across cycles. The number of completed chemotherapy cycles was recorded as a clinical variable.

### 3.3 Sociodemographic and clinical variables

Details concerning sociodemographic profiles encompassed place of residence, occupational status, monthly earnings, educational attainment, marital status, gender, and age. Clinical characteristics included cancer site (colon or rectum), disease stage, pathological type, number of chemotherapy cycles completed, chemotherapy regimen, time since diagnosis, and presence of comorbidities.

These variables were collected for two reasons. First, sociodemographic and clinical characteristics have been shown to influence patients' decision-making experiences, psychological adaptation, and the development of treatment-related regret; therefore, they serve as important contextual factors for interpreting the study results [26]. Second, these characteristics may act as potential covariates or confounders in the relationships among social support, decision self-efficacy, and decision regret. Including them allows for a more accurate description of the sample and facilitates comparison with findings from previous studies.

### 3.4 Sample size

G*Power 3.1 facilitated prior computation of the required sample magnitude, based on parameters encompassing a two-tailed α level of 0.05, moderate effect magnitude ($F^2 = 0.15$), statistical power ($1 - β$) at 0.90, and incorporation of 22 predictors, which established a foundational threshold of 198 subjects. Incorporation of a 20% allowance for nonvalid replies elevated revised baseline to 238. The investigation ultimately incorporated 243 enrollees.

To further ensure the adequacy of the sample for SEM. Consistent with widely accepted SEM recommendations suggesting at least 200 participants for stable parameter estimation in models of comparable complexity, our final sample exceeded this empirical benchmark. Moreover, a post-hoc power analysis was conducted using the observed path coefficients from the validated mediation model. Using the achieved sample size (N = 243), the analysis examined the ability to detect the indirect effect of decision self-efficacy. The resulting statistical power exceeded the commonly accepted threshold of 0.80, indicating that the study was sufficiently powered to detect the hypothesized mediation effect and supporting the robustness of the SEM findings [27].

## 4 Measures

### 4.1 Decision regret

The Three-Dimensional Decision Regret Scale, formulated by van Tol-Geerdink et al. in 2016 [28], underwent translation, cultural adaptation and revision into Chinese by Wang et al [29], serving as the primary instrument in this investigation. Evaluation of remorse over choices in individuals with malignancies constitutes its core purpose, incorporating outcome regret, process regret and option regret as its foundational aspects. Responses occur via a five-point Likert format, spanning 1 ("strongly disagree") through 5 ("strongly agree"), wherein entries 1, 3, 4, 7, 8, 10, 13, 14, and 17 demand reversal in scoring. Aggregate values extend between 18 and 90, such that elevated totals signify intensified remorse over choices. Within this inquiry, robust psychometric properties emerged for the instrument, manifested through Cronbach's α values of 0.905, 0.917, and 0.861 across its three components, in sequence..

### 4.2 Social support

Xiao [30] formulated the Social Support Rating Scale (SSRS),incorporating 10 entries distributed across three domains: support utilization (3 entries), objective support (3 entries), and subjective support (4 entries). Components 1–4 and 8–10 utilize single selection formats, each yielding scores between 1 and 4. Five subsidiary elements populate component 5, where four alternatives per element carry weights from 1 to 4, and summation of these yields the overall for that section. Multi-selection structures define components 6–7 assigning 0 to alternative "A" while alternative "B" receives valuation tied to the quantity of subsidiaries endorsed. Aggregates span 12–66, assessed against benchmarks: (1) <33 denotes poor;

(2) 33–45 indicates moderate; (3) >45 signifies good. Greater aggregates reflect enhanced communal aid perception. For this inquiry, the instrument displayed a Cronbach's α value of the scale was 0.922.

### 4.3 Decision self-efficacy

The confidence of patients in making effective health-related decisions was assessed using the Decision Self-Efficacy Scale, which was originally developed by Bunn and O'Connor [31] and later translated and culturally into Chinese by Wang et al [14]. This scale is a unidimensional scale instrument comprising 11 items. A 5-point Likert format governs responses to each component, spanning 0 ("not at all confident") up 4 ("very confident"). Multiplication of the average across those 11 components by 25 produces an aggregate value between 0 and 100, where larger aggregates signify amplified decision self-efficacy. This effort documented a Cronbach's α of 0.969 for the instrument.

### 4.4 Ethical consideration

This study was conducted in accordance with the Declaration of Helsinki and was approved by the Medical Ethics Committee of the participating hospital (approval number: KY2025118). All participants provided written informed consent prior to data collection. Research staff explained the study purpose, procedures, potential risks, confidentiality protections, and voluntary nature of participation before obtaining signatures from participants.

All respondents were adults (≥18 years), with individuals below that threshold entirely omitted; consequently, endorsement by parents or legal custodians emerged as superfluous. Acquisition of documented approval from each enrollee precluded any dispensation of this stipulation by the institutional review board. Surveys underwent fulfillment in a manner devoid of personal disclosure, alongside the absence of any traceable particulars.

## 5 Statistical methods

Data were analyzed using SPSS 27.0 and AMOS 29.0 Descriptive statistics were used to summarize participants' socio-demographic characteristics and the levels of social support, decision self-efficacy, and decision regret. Continuous variables were reported as means, standard deviations, and ranges, whereas categorical variables were described using frequencies and percentages. Normality was assessed using the Shapiro–Wilk test, which indicated that decision regret scores were not normally distributed. Consequently, group comparisons were conducted using the Mann–Whitney U test for dichotomous variables and the Kruskal–Wallis H test for variables with more than two categories, followed by appropriate post hoc tests. Associations between decision regret and continuous variables were examined using with Spearman's rank correlation coefficients, with statistical significance set at $p < 0.05$ (two-tailed). Correlation strength was categorized as weak (0.00–0.30), moderate (0.40–0.60), or strong (0.70–1.00) [32].Structural equation modeling (SEM) was performed in AMOS 29.0 to test the mediating effect of decision self-efficacy. Because the data did not meet the assumption of multivariate normality, maximum likelihood estimation with Bollen–Stine bootstrap correction (5000 resamples) was applied to obtain robust parameter estimates and model fit indices. The significance of indirect effects was evaluated using bias-corrected bootstrap confidence intervals (95%CIs). with mediation considered significant when zero was not included in the interval.Model fit was considered acceptable the following criteria were met: $\chi^2/df < 3$, NFI and CFI ≥ 0.95, RMSEA < 0.08 [33], and SRMR < 0.09 [34]. The final model demonstrated good fit to the data ($\chi^2/df = 1.569$, NFI = 0.974, CFI = 0.990, RMSEA = 0.048).

## 6 Results

### 6.1 Descriptive statistics

Table 1 presents the descriptive statistics of the study participants. The mean age of cancer patients was 62.11 years (SD = 15.41), with 81.1% being male and 79.4% residing in urban areas. Regarding disease stage, stage III was the most

**Table 1. Descriptive characteristics of the study sample (N = 243).**

| Variables | Groups | N (%) | | Decision Regret (M ± SD) | U/H | P | Post hoc‖ |
|---|---|---|---|---|---|---|---|
| Tumor Status | I | 5 (2.1%) | | (47.80 ± 16.39) | 0.364§ | 0.948 | |
| | II | 68 (28.0%) | | (52.44 ± 13.17) | | | |
| | III | 148 (60.9%) | | (52.56 ± 13.68) | | | |
| | IV | 22 (9.1%) | | (53.55 ± 11.60) | | | |
| Chemotherapy Cycles | > 6 | 64 (26.3%) | | | −0.487‡ | 0.627 | |
| | 1-6 | 179 (72.7%) | | | | | |
| Age | | | (62.11 ± 15.41) | | 0† | 0.998 | |
| Gender | Male | 197 (81.1%) | | (52.20 ± 13.93) | −0.565‡ | 0.57 | |
| | Female | 46 (18.9%) | | (53.86 ± 10.85) | | | |
| Education | Junior High or Below | 55 (22.6%) | | (60.76 ± 10.58) | 53.699 | 0.01 | 1 > 2***, 1 > 3***, 2 > 3*** |
| | High School or Equivalent | 102 (42.0%) | | (54.80 ± 7.19) | | | |
| | College degree or above | 86 (35.4%) | | (45.41 ± 10.56) | | | |
| Marital status | Single | 1 (0.7%) | | −56 | 0.505§ | 0.918 | |
| | Married | 189 (77.8%) | | (52.35 ± 13.34) | | | |
| | Divorced | 17 (7.0%) | | (55.94 ± 5.18) | | | |
| | Widowed | 36 (14.8%) | | (51.63 ± 13.27) | | | |
| Residence | Rural | 50 (20.6%) | | (54.08 ± 7.70) | −0.054‡ | 0.957 | |
| | Urban | 193 (79.4%) | | (52.11 ± 14.50) | | | |
| Ethnicity | Han | 216 (88.9%) | | (51.96 ± 13.18) | −1.387‡ | 0.166 | |
| | Mongol | 23 (9.5%) | | (54.95 ± 14.86) | | | |
| Type of Disease | Colon cancer | 170 (70.0%) | | (52.30 ± 13.11) | −0.289‡ | 0.773 | |
| | Rectal cancer | 73 (30.0%) | | (53.01 ± 11.64) | | | |
| Monthly income (yuan) | <3000 | 60 (24.7%) | | (61.86 ± 8.47) | 48.13§ | 0.01 | 1 > 3***, 2 > 3***, 1 > 2* |
| | 3000-5000 | 70 (28.8%) | | (55.95 ± 6.58) | | | |
| | >5000 | 113 (46.5%) | | (45.41 ± 15.02) | | | |
| Living situation | Living alone | 13 (5.3%) | | (54.08 ± 15.92) | 2.779§ | 0.249 | |
| | Living with family | 194 (79.8%) | | (51.56 ± 14.08) | | | |
| | Other | 36 (14.8%) | | (57.11 ± 13.40) | | | |
| Main sources of medical expenses | Self-pay | 25 (10.3%) | | (51.68 ± 10.70) | 2.351§ | 0.309 | |
| | New Rural Cooperative Medical Scheme | 118 (48.6%) | | (54.70 ± 9.40) | | | |
| | Urban Employee/Resident Basic Medical Insurance | 100 (41.2%) | | (50.16 ± 11.18) | | | |
| Occupation | Unemployed | 34 (14.0%) | | (62.59 ± 11.02) | 43.269§ | 0.01 | 1 > 4***, 1 > 3***, 1 > 2*, 2 > 4*, 2 > 5* |
| | Farmer | 75 (31.0%) | | (58.83 ± 8.27) | | | |
| | Public institution staff | 55 (22.6%) | | (48.43 ± 13.87) | | | |
| | Self-employed | 50 (20.2%) | | (48.28 ± 13.04) | | | |
| | Retired | 29 (12.2%) | | (46.96 ± 12.15) | | | |
| Family history of cancer | No | 219 (90.1%) | | (52.12 ± 13.47) | −0.181‡ | 0.856 | |
| | Yes | 24 (9.9%) | | (53.37 ± 13.00) | | | |
| Stoma | No | 216 (88.9%) | | (52.15 ± 13.59) | −1.657‡ | 0.098 | |
| | Yes | 27 (11.1%) | | (55.48 ± 11.59) | | | |

*(Continued)*

**Table 1.** (Continued)

| Variables | Groups | N (%) | | Decision Regret (*M±SD*) | *U/H* | *P* | Post hoc‖ |
|---|---|---|---|---|---|---|---|
| Chemotherapy regimen | FOLFOX | 43 (17.7%) | | (51.579±13.83) | 2.218§ | 0.528 | |
| | XELOX | 120 (49.4%) | | (51.57±14.22) | | | |
| | FOLFIRI | 38 (15.6%) | | (53.58±12.49) | | | |
| | Others | 42 (17.3%) | | (55.02±11.18) | | | |

† Spearman's correlation analysis.

‡ Mann-Whitney U test.

§ Kruskal-Wallis H test.

‖Post hoc analyses adjusted by Bonferroni test; only significant correlations are listed.

*** P<0.001.

** P<0.01.

*P<0.05.

common (60.9%). In terms of occupation, farmers accounted for the largest proportion (31.0%). For marital status, the majority of patients were married (77.8%). Table 2 shows the mean scores for social support, decision self-efficacy, and decision regret, which were 37.63±9.42, 27.64±8.87, and 52.52±13.40, respectively.

Statistically significant differences in decision regret were observed across education level (H=53.699, P<0.001), monthly income (H=48.130, p<0.001), and occupation (H=43.269, P<0.001). Post hoc analyses indicated that patients with lower educational attainment reported significantly higher decision regret compared with those with higher education (P<0.05). Similarly, patients with lower monthly income reported greater decision regret compared with those with higher income (P<0.05). In addition, unemployed patients reported significantly higher decision regret compared with employees in public institutions and retired individuals (P<0.05).

## 6.2 Correlation analysis of social support, decision self-efficacy, and decision regret

Table 2 illustrates meaningful interconnections tying decision regret, social support, and decision self-efficacy together. The connection between social support and decision self-efficacy took a favorable direction (r=0.344, p<0.001), whereas the bond to decision regret assumed an unfavorable slant (r=−0.450, p<0.001). In parallel fashion, decision self-efficacy linked inversely to decision regret (r=−0.581, p<0.001).

**Table 2. Scores of social support, decision self-efficacy and decision regret of colorectal cancer patients(n=243).**

| | M | SD | Decision Regret | Process regret | Option regret | Outcome regret | Social support | Objective support | Subjective support | Utilisation of support | Decision-self-efficacy |
|---|---|---|---|---|---|---|---|---|---|---|---|
| Decision Regret | 52.52 | 13.40 | 1 | | | | | | | | |
| Process regret | 20.65 | 5.46 | 0.922** | 1 | | | | | | | |
| Option regret | 20.10 | 5.41 | 0.882** | .739** | 1 | | | | | | |
| Outcome regret | 11.77 | 2.92 | 0.841** | .726** | 0.765** | 1 | | | | | |
| Social support | 37.63 | 9.42 | −0.450** | −0.428** | −0.440** | −0.439** | 1 | | | | |
| Objective support | 12.09 | 2.52 | −0.402** | −0.392** | −0.369** | −0.400** | 0.831** | 1 | | | |
| Subjective support | 18.13 | 5.39 | −0.458** | −0.440** | −0.445** | −0.456** | 0.926** | 0.685** | 1 | | |
| Utilisation of support | 7.41 | 2.08 | −0.510** | −0.496** | −0.506** | −0.425** | 0.787** | 0.613** | 0.674** | 1 | |
| Decision-self-efficacy | 27.64 | 8.87 | −0.581** | −0.589** | −0.592** | −0.548** | 0.344** | 0.326** | 0.329** | 0.405** | 1 |

Abbreviations: M, mean; SD, standard deviation.** p < 0.01.

### 6.3 Mediating effect of decision self-efficacy

Investigation into the intermediary function served by decision self-efficacy within the association linking social support to decision regret prompted assembly of an SEM via AMOS. To control for potential confounding effects, education level, monthly personal income, and occupation were included as covariates in the model. These covariates were allowed to correlate with the independent variable (social support) and were specified to exert direct effects on both the mediator and the outcome variable. The significance of the mediating effect was tested using a bias-corrected percentile bootstrap method with 5,000 resamples. Alignment metrics conveyed that the proposed trajectory configuration aligned adequately against observations ($\chi^2$/df = 1.569, p = 0.020, CFI = 0.990, NFI = 0.974, RMSEA = 0.048). Outcomes revealed partial channeling by decision self-efficacy across the pathway connecting social support with decision regret. Pathway influence indirectly through decision self-efficacy registered at –0.273, paired with a deviation-compensated 95% CI ranging from –0.680 through –0.467 (p < 0.01). Cumulative influence from social support upon decision regret routed via decision self-efficacy, equaled –0.583 (p < 0.01). Robustness probes under varied conditions upheld the core observations without alteration. Elaborate evaluations from the configuration scrutiny reside within Table 3.

## 7 Discussion

### 7.1 Summary of main results

In this study, the mean decision regret score among CRC patients was 52.52 (SD = 13.40), indicating a moderate level of regret. Decision regret levels varied by across education level, occupation, and monthly personal income. Additionally, higher levels of social support were associated with lower decision regret, and this relationship was partially mediated by decision self-efficacy, which confirmed our hypothesis.

### 7.2 Sociodemographic variables associated with decision regret

This study found a significant association between decision regret and income. Consistent with previous research [26,35], patients with poorer economic conditions were more likely to experience regret, primarily due to limited treatment options and restricted access to information [36].Moreover, compared to patients with other types of cancer, those with CRC cancer tend to be more passive in treatment decision-making [37]. Given that chemotherapy often involves prolonged treatment cycles, high costs, and significant side effects, low-income patients are more likely to experience hesitation and post-decision reflection under the dual burden of financial and life stress, which further exacerbates decision regret [38].

In addition, decision regret was also significantly associated with occupation. Our findings indicated that unemployed patients reported significantly higher levels of regret compared to employees in public institutions and retirees. This may be related to the reduced financial resources and diminished social support often associated with unemployment. Unemployed patients frequently worry about the economic burden of treatment on their families, the potential decline in quality

**Table 3. Mediating Effect of Decision-self-efficacy on the Relationship Between Social Support and Decision Regret.**

| | Effect of X on M | Effect of M on Y | Direct effect | Indirect effect | Total effect | Effect size | Bootstrapping |
|---|---|---|---|---|---|---|---|
| | | | | | | | (BC 95% CI) |
| Social support→Decision-self-efficacy→Decision Regret | 0.471 | −0.579 | −0.31 | −0.273 | −0.583 | 47.24% | [-0.364, -0.196] |
| | CI, confidence interval; M, Decision self-efficacy; X,Social support; Y,Decision Regret | | | | | | |
| | * Adjusted for monthly income,education, and occupation | | | | | | |

of life, and even the possibility of incurring debt, making them more vulnerable to self-doubt and regret after treatment [39]. Therefore, clinical practice should address the psychological burden of decision-making among low-income and unemployed patients. Providing financial support, cost counseling, and psychological interventions, in combination with strengthening shared decision-making, may help reduce decision regret, improve adherence, and enhance quality of life.

In this study, we found that decision regret among CRC chemotherapy patients significantly decreased with increasing educational level. Specifically, patients with higher education levels reported lower levels of decision regret during the treatment decision-making process. This finding may be attributed to higher health literacy, which enables individuals to better understand complex chemotherapy regimens and more effectively express their personal values, thereby reducing uncertainty and regret [40,41]. In contrast, previous studies have reported differing results. Step et al. [42] have suggested that both patients with lower education levels (associate degree, high school or less) and those with higher education levels (master's degree or above) are more likely to experience decision regret, whereas those with intermediate education (college) report lower levels of regret, indicating a curvilinear relationship. This phenomenon may be explained by the tendency of highly educated patients to engage in excessive reflection when treatment outcomes are unsatisfactory, thus exacerbating feelings of regret [43].

## 7.3 The mediating role of decision self-efficacy

Central among theoretical advancements advanced by this investigation lies proof of partial channeling through decision self-efficacy along the pathway joining social support with decision regret encompassing 47% of overall influence. Evidence of this sort lends observational backing to Social Cognitive Theory amid selections for cancer therapies, while broadening earlier inquiries centered largely on prostate and breast malignancy groups [44,45]. Social support appears to exert its protective effect through two distinct pathways. First, consistent with the stress-buffering model, social support provides emotional reassurance and practical assistance, directly alleviating the psychological burden associated with complex CRC treatment decisions. These forms of support may reduce counterfactual thinking and feelings of isolation, thereby decreasing the likelihood of regret [9,46,47]. Second, and more importantly in quantitative terms, social support enhances patients' confidence in their ability to participate meaningfully in treatment decisions. Family members who accompany patients to consultations, help interpret medical information, and offer encouragement directly strengthen patients' perceived control, an essential antecedent of decision self-efficacy [19,48]. In turn, patients with higher decision self-efficacy engage more actively in shared decision-making, ask more questions, and feel a greater sense of ownership over the final choice [23,24]. Even when outcomes are suboptimal, these patients are better equipped to reconcile their decisions with reality, resulting in significantly less regret.

The mediating role of decision self-efficacy has important clinical implications. Interventions that focus solely on enhancing external social support are likely to lead to only limited and potentially transient reductions in decision regret unless patients' internal decision-making competence is also strengthened. This is particularly relevant in colorectal cancer, where treatment decisions typically require patients to weigh survival benefits against substantial, often permanent, quality-of-life consequences [49]. For example, chemotherapy and stoma formation frequently result in changes to physical appearance and bowel function, directly impairing quality of life and heightening patients' sensitivity to social interactions [50]. Patients with high decision self-efficacy tend to engage more actively and confidently with the healthcare team. They are better able to articulate their values, concerns, and preferences, fostering mutual trust, promoting transparent and open shared decision-making, and reducing uncertainty during the decision process [51,52]. This proactive engagement not only mitigates decisional conflict at the time of choice but also significantly lowers the risk of subsequent regret arising from perceived information deficits or lack of personal involvement [52,53].

The study found that decision self-efficacy exhibited different correlation patterns with various types of decision regret. It was more strongly correlated with process regret and option regret, but relatively weakly correlated with outcome regret. This may be due to the fact that treatment outcomes, such as the adverse effects of chemotherapy, are often beyond

the patient's control, which is consistent with previous studies [54]. Even if patients possess strong decision self-efficacy, insufficient health literacy can make it difficult for them to fully understand CRC chemotherapy regimens and subsequent treatments, leading to dissatisfaction or regret regarding treatment outcomes [55].

Further analysis revealed that the utilization of social support was most strongly correlated with decision self-efficacy, emphasizing the critical role of patients' proactive engagement with available support resources in enhancing self-efficacy. In the context of CRC and chemotherapy, this is particularly important, as the complex and prolonged nature of treatment decisions—such as managing fatigue, gastrointestinal distress, and immune suppression, adds an additional layer of uncertainty. These factors make patients more vulnerable to decision regret [4]. These findings highlight the need for healthcare providers to not only focus on strengthening patients' external support networks but also on fostering their internal psychological resources, particularly self-efficacy. This is crucial in helping patients navigate complex, high-stakes treatment decisions, ultimately reducing decision regret.

## 8 Limitations

Several constraints characterize this inquiry. Primarily, adoption of a cross-sectional framework obstructs derivation of causality and curtails capacity to delineate chronological linkages connecting decision regret, decision self-efficacy, and social support, all the while fostering susceptibility to retrospective distortion. Next, recruitment occurred solely from one advanced-care facility located in Inner Mongolia, a circumstance compounded by reliance on opportunistic enrollment that could curtail applicability of outcomes beyond this locale and engender preferential enrollment distortions, given that individuals under tertiary oversight typically encounter amplified availability of aids and backing relative to counterparts in alternative therapeutic contexts. Even so, substantive initial insights emerge from this work into mental processes shaping choices for chemotherapy among those afflicted with CRC. Future research should employ multi-center, longitudinal designs to strengthen external validity and clarify the temporal pathways among the study variables.

## 9 Implications for nursing practice

This study revealed that CRC patients with lower levels of social support and decision self-efficacy are more likely to experience higher levels of decision regret. Based on the results, nursing interventions should be tailored to enhance patients' confidence in participating in treatment decisions and to improve their access to social resources. Specifically, for patients with insufficient self-efficacy, more structured educational programs and individualized counseling are needed to improve their decision-making competence. For patients with limited social support, nurses may focus on facilitating peer support networks and strengthening family involvement in the decision-making process. Additionally, given the collectivist orientation of Chinese culture, the involvement of family members in treatment decisions plays a critical role in reducing patients' regret, emphasizing that cancer care should be family-centered. Therefore, a comprehensive nursing support system including patient decision aids, family communication skills training, and psychosocial interventions should be developed and integrated into CRC care. Based on these findings, further interventional studies are needed to validate the effectiveness of such nursing strategies in reducing decision regret and improving long-term psychological outcomes among patients.

## 10 Conclusion

The level of decision regret among CRC patients was significantly associated with occupation, monthly income, and educational attainment. Additionally, both social support and decision self-efficacy were negatively associated with decision regret, with decision self-efficacy mediating the effect of social support on decision regret. These findings suggest that, in the context of CRC care, greater emphasis should be placed on developing targeted interventions and strengthening social support systems to effectively alleviate patients' decision regret.

## Supporting information

**S1 Dataset. Raw Data Used for Statistical Analyses.**
(XLSX)

## Author contributions

**Conceptualization:** Fang Wu.

**Data curation:** Zhichao Duan, Ying Li.

**Investigation:** Zhichao Duan, Ying Li.

**Software:** Zhichao Duan, Gege Ren.

**Supervision:** Fang Wu.

**Writing – original draft:** Zhichao Duan, Miao Tan.

**Writing – review & editing:** Zhichao Duan.

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
