## [Decision Letter · Decision Letter 0]

17 Nov 2025

Dear Dr. Wu,

Thank you for submitting your manuscript to PLOS ONE. After careful consideration, we feel that it has merit but does not fully meet PLOS ONE’s publication criteria as it currently stands. Therefore, we invite you to submit a revised version of the manuscript that addresses the points raised during the review process.

We look forward to receiving your revised manuscript.

Kind regards,

Jamshed Akhtar, MBBS, FCSP, FRCS, MHPE, FACS, M Bioethics

Academic Editor

PLOS ONE

Journal Requirements:

3. Thank you for submitting the above manuscript to PLOS ONE. During our internal evaluation of the manuscript, we found significant text overlap between your submission and previous work in the [introduction, conclusion, etc.].

Please revise the manuscript to rephrase the duplicated text, cite your sources, and provide details as to how the current manuscript advances on previous work. Please note that further consideration is dependent on the submission of a manuscript that addresses these concerns about the overlap in text with published work.

[If the overlap is with the authors’ own works: Moreover, upon submission, authors must confirm that the manuscript, or any related manuscript, is not currently under consideration or accepted elsewhere. If related work has been submitted to PLOS ONE or elsewhere, authors must include a copy with the submitted article. Reviewers will be asked to comment on the overlap between related submissions (http://journals.plos.org/plosone/s/submission-guidelines#loc-related-manuscripts).]

We will carefully review your manuscript upon resubmission and further consideration of the manuscript is dependent on the text overlap being addressed in full. Please ensure that your revision is thorough as failure to address the concerns to our satisfaction may result in your submission not being considered further

4. Please include captions for your Supporting Information files at the end of your manuscript, and update any in-text citations to match accordingly. Please see our Supporting Information guidelines for more information: http://journals.plos.org/plosone/s/supporting-information .

Additional Editor Comments (if provided):

Observations of reviewers are attached for you to reply.

Reviewers' comments:

Reviewer's Responses to Questions

**Comments to the Author**

1. Is the manuscript technically sound, and do the data support the conclusions?

Reviewer #1: Yes

Reviewer #2: Yes

2. Has the statistical analysis been performed appropriately and rigorously?

Reviewer #1: Yes

Reviewer #2: Yes

3. Have the authors made all data underlying the findings in their manuscript fully available?

Reviewer #1: Yes

Reviewer #2: Yes

4. Is the manuscript presented in an intelligible fashion and written in standard English?

Reviewer #1: Yes

Reviewer #2: Yes

Reviewer #1: Dear Editor,

Thank you for the opportunity to review the manuscript titled “Relationships Among Social Support, Decision Self-efficacy, and Decision Regret in colorectal Chemotherapy cancer patients: A Mediating Model”. This research explores the mediating role of Decision Self-efficacy in the association between Social Support and Decision Regret. This study tackles a valuable topic, but the manuscript in its current form lacks sufficient detail and clarity in key sections. My specific comments below are intended to help the authors address these issues.

Introduction

- The manuscript would be strengthened by a more explicit theoretical justification for the proposed relationships. While the variables are introduced, the logical argument connecting Social Support and Decision Self-Efficacy to Decision Regret is implicit. Clearly stating the hypothesized pathways, with support from relevant theory, will make a more compelling case for the study's importance.

Method

- The date provided on page X should be written in a more formal academic style. Please revise it from 29/07/2025 to 02/10/2025 to a standard format July to October 2025.

- The exclusion criteria listed as “(1) presence of psychiatric disorders; (2) cognitive impairment; (3) coexistence of other malignancies; (4) severe hepatic or renal dysfunction;” are direct logical opposites of the inclusion criteria “(1) pathologically confirmed CRC; (2) completion of at least one cycle of chemotherapy; (3) age ≥18 years; (4) clear consciousness; (5) ability to communicate normally; and (6) complete clinical research data.”. To avoid redundancy and improve the clarity of the methodology, the repeated exclusion criteria should be removed.

- The sample size justification appears to be based on a power analysis for a simpler statistical test. However, given that the primary analysis is Structural Equation Modeling (SEM), the sample size calculation should be refined. It is recommended to perform a power analysis specific to SEM, considering the complexity of the proposed model (e.g., the number of latent variables and estimated parameters) to ensure the study is adequately powered to detect the hypothesized effects.

- A detailed description of the sociodemographic and clinical variables collected (e.g., age, gender, education, cancer type, chemotherapy regimen, disease stage). The rationale for including these specific variables should be briefly explained.

- The statistical section acknowledges non-normal data but does not address its implications for the SEM analysis. Normality is a key assumption for the standard Maximum Likelihood estimator. The authors must either:

a) Provide evidence that the deviation from normality is not severe enough to bias the results (e.g., by reporting skewness and kurtosis statistics), or

b) Re-run the analysis using an estimation method robust to non-normality (e.g., MLR, MLM, or bootstrapping) and report those results instead. The current findings cannot be considered reliable until this issue is resolved.

Result

- The correlations for the social support domains are reported separately. To strengthen the manuscript, please provide a rationale for this analytical choice, discussing whether specific domains were expected to have distinct relationships with the outcome variables or combine all of its domain.

- The results section for the mediating effect of decision self-efficacy reports a significant indirect effect. However, as this mediation was not included in the study's stated hypotheses, this finding appears to be exploratory.

Discussion

- The discussion section should focus on interpreting the results and their implications, rather than reiterating the methods. Please remove the description of the analytical method from the discussion.

- The discussion would benefit from a rebalancing of its focus. While the direct association between social support and decision regret is noted, the primary theoretical contribution of this study is the mediating role of decision self-efficacy. I recommend condensing the discussion of the direct association and significantly expanding the interpretation of the mediation finding.

Reviewer #2: Summary of research.

The title of the study is clearly reflecting the study’s aims. It effectively describes the significance of Colorectal cancer as a public health issue and it logically incorporates its key concepts; decision regret, social support, decision self- efficacy. The hypothesis is stated clearly and aligned with the rationale of the study. Conclusion is concise and actionable. The study is highly relevant and clinically significant to the existing scenario.

Overall Assessment:

Strengths:

Clear theoretical foundation & hypothesis. Large sample size, High internal reliability (Cronbach’s alpha >0.86). Hypotheses >2 align with mediation model.

Methodology & Results are clear and summarized accurately. The choice of statistical tests is well justified.

Weaknesses:

Cross sectional design is the most significant limitation.If the study is cross sectional , its conclusions must be displayed with caution, calling it a preliminary study which cannot be generalizable.

Minor typos/incomplete sentences. e.g. “accompanied by (3) missing text -----

“Lee & Bryant-Lukosius (14) “ citation mismatch with text.

Cited study by “Temitope G” lacks full reference

Hypothesis needs rephrasing as it is hard to comprehend and is weakly worded, e.g. “ Patient’s social support and self efficacy may account for

There is no mention of measurement timing of different variables checked during the course of chemotherapy – variable cycles?

Convenience sampling is fine but risks selection bias (e.g. patients in tertiary care hospitals may have better access/support)

Future directions need to be added with emphasis on multi center longitudinal study.

**Do you want your identity to be public for this peer review?** For information about this choice, including consent withdrawal, please see our Privacy Policy

Reviewer #1: No

Reviewer #2: No

---

## [Author Response · Author response to Decision Letter 1]

4 Dec 2025

Thank you very much for the insightful and constructive comments provided by the editor and reviewers. We greatly appreciate the time and expertise invested in evaluating our manuscript. We have carefully considered all suggestions and have revised the manuscript accordingly. Below, we provide a detailed, point-by-point response to each comment. All corresponding changes have been incorporated into the revised manuscript, with revisions clearly marked for ease of review.

We sincerely appreciate the opportunity to improve our work and believe that the revisions have substantially strengthened the clarity, rigor, and contribution of the manuscript.

---

## [Editor Report · Decision Letter 1]

14 Dec 2025

Relationships Among Social Support, Decision Self-efficacy, and Decision Regret in colorectal Chemotherapy cancer patients: A Mediating Model

PONE-D-25-53912R1

Dear Dr. Wu,

We’re pleased to inform you that your manuscript has been judged scientifically suitable for publication and will be formally accepted for publication once it meets all outstanding technical requirements.

Kind regards,

Jamshed Akhtar, MBBS, FCSP, FRCS, MHPE, FACS, M Bioethics

Academic Editor

PLOS One
---

## [Editor Report · Acceptance letter]

PONE-D-25-53912R1

PLOS One

Dear Dr. Wu,

I'm pleased to inform you that your manuscript has been deemed suitable for publication in PLOS One. Congratulations! Your manuscript is now being handed over to our production team.

Kind regards,

on behalf of

Dr. Jamshed Akhtar

Academic Editor

PLOS One